# A Ranking Game for Imitation Learning

## Abstract

We propose a new framework for imitation learning—treating imitation as a *two-player ranking-based game* between a policy and a reward. In this game, the reward agent learns to satisfy pairwise performance rankings between behaviors, while the policy agent learns to maximize this reward. In imitation learning, near-optimal expert data can be difficult to obtain, and even in the limit of infinite data cannot imply a total ordering over trajectories as preferences can. On the other hand, learning from preferences alone is challenging as a large number of preferences are required to infer a high-dimensional reward function, though preference data is typically much easier to collect than expert demonstrations. The classical inverse reinforcement learning (IRL) formulation learns from expert demonstrations but provides no mechanism to incorporate learning from offline preferences and vice versa. We instantiate the proposed ranking-game framework with a novel ranking loss giving an algorithm that can simultaneously learn from expert demonstrations and preferences, gaining the advantages of both modalities. Our experiments show that the proposed method achieves state-of-the-art sample efficiency and can solve previously unsolvable tasks in the Learning from Observation (LfO) setting.

## 1 Introduction

Reinforcement learning relies on environmental reward feedback to learn meaningful behaviors. Reward specification is a hard problem [39], thus motivating imitation learning (IL) as a technique to bypass reward specification and learn from expert data, often via Inverse Reinforcement Learning (IRL) techniques. Learning from expert observations (imitation learning) alone can require efficient exploration when the expert actions are unavailable as in LfO [36]. Incorporating preferences over potentially suboptimal trajectories for reward learning can help reduce the exploration burden by regularizing the reward function and providing effective guidance for policy optimization. Previous literature in learning from preferences either assumes no environment interaction [10, 9] or assumes an active query framework with a restricted reward class [47]. The classical IRL formulation suffers from two issues: (1) Learning from expert demonstrations and learning from preferences/rankings provide complementary advantages for increasing learning efficiency [30, 47]; however, existing IRL methods that learn from expert demonstrations provide no mechanisms to incorporate offline preferences and vice versa. (2) Optimization is difficult, making learning sample inefficient [5, 28] due to the adversarial min-max game.

Our primary contribution is an algorithmic framework casting imitation learning as a ranking game which addresses both of the above issues in IRL. This framework treats imitation as a ranking game between two agents: a reward agent and a policy agent—the reward agent learns to satisfy pairwise performance rankings between different *behaviors*

represented as state-action or state visitations, while the policy agent maximizes its performance under the learned reward function. The ranking game is detailed in Figure 1 and is specified by three components: (1) The dataset of pairwise behavior rankings, (2) A ranking loss function, and (3) An optimization strategy. This game encompasses a large subset of both inverse reinforcement learning (IRL) methods and methods which learn from suboptimal offline preferences. Popular IRL methods such as GAIL, AIRL, $f$-MAX [28, 22, 34] are instantiations of this ranking game in which rankings are given only between the learning agent and the expert, and a gradient descent ascent (GDA) optimization strategy is used with a ranking loss that maximizes the performance gap between the behavior rankings.

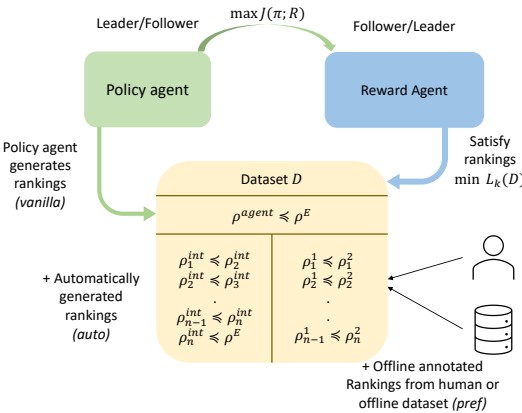

The ranking loss used by the prior IRL approaches is specific to the comparison of optimal (expert) vs. suboptimal (agent) data, and precludes incorporation of comparisons among suboptimal behaviors. In this work, we instantiate the ranking game by proposing a new ranking loss ($L_k$) that facilitates incorporation of rankings over suboptimal trajectories for reward learning. Our theoretical analysis reveals that the proposed ranking loss results in a bounded performance gap with the expert that depends on a controllable hyperparameter. Our ranking loss can also ease policy optimization by supporting data augmentation to make the reward landscape smooth and allowing control over the learned reward scale. Finally, viewing our ranking game in the Stackelberg game framework (see Section 3)—an efficient setup for solving general-sum games—we obtain two algorithms with complementary benefits in non-stationary environments depending on which agent is set to be the leader.

Figure 1: `rank-game`: The Policy agent maximizes the reward function by interacting with the environment. The Reward agent satisfies a set of behavior rankings obtained from various sources: generated by the policy agent (vanilla), automatically generated (auto), or offline annotated rankings obtained from a human or offline dataset (pref). Treating this game in the Stackelberg framework leads to either Policy being a leader and Reward being a follower, or vice-versa.

In summary, this paper formulates a new framework `rank-game` for imitation learning that allows us to view learning from preferences and demonstrations under a unified perspective. We instantiate the framework with a principled ranking loss that can naturally incorporate rankings provided by diverse sources. Finally, by incorporating additional rankings—auto-generated or offline—our method: (a) outperforms state-of-the-art methods for imitation learning in several MuJoCo simulated domains by a significant margin and (b) solves complex tasks like imitating to reorient a pen with dextrous manipulation using only a few observation trajectories that none of the previous LfO baselines can solve.

## 2 Related Work

Imitation learning methods are broadly divided into two categories: Behavioral cloning [48, 54] and Inverse Reinforcement Learning (IRL) [44, 1, 72, 18, 20, 28, 22]. Our work focuses on developing a new framework in the setting of IRL through the lens of ranking. Table 1 shows a comparison of the proposed `rank-game` method to prior works.

**Classical Imitation Game for IRL**: The classical imitation game for IRL aims to solve the adversarial *min-max* problem of finding a policy that minimizes the worst-case performance gap between the agent and the expert. A number of previous works [22, 60, 34] have focused on analyzing the properties of this *min-max* game and its relation to divergence minimization. Under some additional regularization, this *min-max* objective can be understood as minimizing a certain $f$-divergence [28, 22, 34] between the agent and expert state-action visitation. More recently, [60] showed that all forms of imitation learning (BC and IRL) can be understood as performing moment matching under differing assumptions. In this work, we present a new perspective on imitation

| IL Method | Offline Preferences | Expert Data | Ranking Loss | Reward Function | Active Human Query |
|---|---|---|---|---|---|
| MaxEntIRL, AdRIL,GAN-GCL, GAIL,$f$-MAX, AIRL | ✗ | LfD | supremum | non-linear | ✗ |
| BCO,GAIfO, DACfO, OPOLO,$f$-IRL | ✗ | LfO | supremum | non-linear | ✗ |
| TREX, DREX | ✓ | ✗ | Bradley-Terry | non-linear | ✗ |
| BREX | ✓ | ✗ | Bradley-Terry | linear | ✗ |
| DemPref | ✓ | LfO/LfD | Bradley-Terry | linear | ✓ |
| Ibarz et al[30] | ✓ | LfD | Bradley-Terry | non-linear | ✓ |
| `rank-game` | ✓ | LfO/LfD | $L_k$ | non-linear | ✗ |

Table 1: A summary of IL methods demonstrating the data modalities they can handle (expert data and/or preferences), the ranking-loss functions they use, the assumptions they make on reward function, and whether they require availability of an external agent to provide preferences during training. We highlight whether a method enables LfD, LfO, or both when it is able to incorporate expert data.

in which the reward function is learned using a dataset of behavior comparisons, generalizing previous IRL methods that learn from expert demonstrations and additionally giving the flexibility to incorporate rankings over suboptimal behaviors.

**Learning from Preferences and Suboptimal Data**: Learning from preferences and suboptimal data is important when expert data is limited or hard to obtain. Preferences [3, 65, 55, 14, 47] have the advantage of providing guidance in situations expert might not get into, and in the limit provides full ordering over trajectories which expert data cannot. A previous line of work [10, 11, 9, 13] has studied this setting and demonstrated that offline rankings over suboptimal behaviors can be effectively leveraged to learn a reward function. [14, 47, 30] studied the question of learning from preferences in the setting when a human is available to provide online preferences[1] (active queries), while [47] additionally assumed the reward to be linear in known features. Our work makes no such assumptions and allows for integrating offline preferences and expert demonstrations under a common framework.

**Learning from Observation** (LfO): LfO is the problem setting of learning from expert observations. This is typically more challenging than the traditional learning from demonstration setting (LfD), because actions taken by the expert are unavailable. LfO is broadly formulated using two objectives: state-next state marginal matching [63, 71, 58] and direct state marginal matching [45, 43]. Some prior works [61, 67, 16] approach LfO by inferring expert actions through a learned inverse dynamics model. These methods assume injective dynamics and suffer from compounding errors when the policy is deployed. A recently proposed method OPOLO [71] derives an upper bound for the LfO objective which enables it to utilize off-policy data and increase sample efficiency. Our method outperforms baselines including OPOLO, by a significant margin.

## 3   Background

We consider a learning agent in a Markov Decision Process (MDP) [49, 59] which can be defined as a tuple: $\mathcal{M} = (\mathcal{S}, \mathcal{A}, P, R, \gamma, \rho_0)$, where $\mathcal{S}$ and $\mathcal{A}$ are the state and action spaces; $P$ is the state transition probability function, with $P(s'|s, a)$ indicating the probability of transitioning from $s$ to $s'$ when taking action $a$; $R : \mathcal{S} \times \mathcal{A} \to \mathbb{R}$ is the reward function bounded in $[0, R_{max}]$; We consider MDPs with infinite horizon, with the discount factor $\gamma \in [0, 1]$, though our results extend to finite horizons as well; $p_0$ is the initial state distribution. We use $\Pi$ and $\mathcal{R}$ to denote the space of policies and reward functions respectively. A reinforcement learning agent aims to find a policy $\pi : \mathcal{S} \to \mathcal{A}$ that maximizes its expected return, $J(R; \pi) = \frac{1}{1-\gamma}\mathbb{E}_{(s,a)\sim\rho^\pi(s,a)}[R(s, a)]$, where $\rho^\pi(s, a)$ is the stationary state-action distribution induced by $\pi$. In imitation learning, we are provided with samples from the state-action visitation of the expert $\rho^{\pi_E}(s, a)$ but the reward function of the expert is unknown. We will use $\rho^E(s, a)$ as a shorthand for $\rho^{\pi_E}(s, a)$.

**Classical Imitation Learning**: The goal of imitation learning is to close the imitation gap $J(R; \pi^E) - J(R; \pi)$ defined with respect to the unknown expert reward function $R$. Several prior works [28, 60, 38, 45] tackle this problem by minimizing the imitation gap on all possible reward hypotheses. This

---

[1]We will use preferences and ranking interchangebly

leads to a zero-sum (min-max) game formulation of imitation learning in which a policy is optimized with respect to the reward function that induces the largest imitation gap:

$$\texttt{imit-game}(\pi) = \arg\min_{\pi \in \Pi} \sup_{f \in \mathcal{R}} \mathbb{E}_{\rho^E(s,a)}[f(s,a)] - \mathbb{E}_{\rho^\pi(s,a)}[f(s,a)]. \tag{1}$$

Here, the imitation gap is upper bounded as follows ($\forall \pi$):

$$J(R; \pi^E) - J(R; \pi) \le \sup_{f \in \mathcal{R}} \mathbb{E}_{\rho^E(s,a)}[f(s,a)] - \mathbb{E}_{\rho^\pi(s,a)}[f(s,a)]. \tag{2}$$

Note that, when the performance gap is maximized between the expert $\pi^E$ and the agent $\pi$, we can observe that the worst-case reward function $f_\pi$ induces a ranking between policy behaviors based on their performance: $\rho^E \succeq \rho^\pi := \mathbb{E}_{\rho^E(s,a)}[f_\pi(s,a)] \ge \mathbb{E}_{\rho^\pi(s,a)}[f_\pi(s,a)], \ \forall \pi$. Therefore, we can regard the above loss function that maximizes the performance gap (Eq. 2) as an instantiation of the ranking-loss. We will refer to the implicit ranking between agent and the expert $\rho^E \succeq \rho^\pi$ as vanilla rankings and this variant of the ranking-loss function as the *supremum-loss*.

**Stackelberg Games**: A Stackelberg game is a general-sum game between two agents where one agent is set to be the leader and the other a follower. The leader in this game optimizes its objective under the assumption that the follower will choose the best response for its own optimization objective. More concretely, assume there are two players $A$ and $B$ with parameters $\theta_A, \theta_B$ and corresponding losses $\mathcal{L}_A(\theta_A, \theta_B)$ and $\mathcal{L}_B(\theta_A, \theta_B)$. A Stackelberg game solves the following bi-level optimization when $A$ is the leader and $B$ is the follower: $\min_{\theta_A} \mathcal{L}_A(\theta_A, \theta_B^*(\theta_A))$ s.t $\theta_B^*(\theta_A) = \arg\min_\theta \mathcal{L}_B(\theta_A, \theta)$. [51] showed that casting model-based RL as an approximate Stackelberg game [6] leads to performance benefits and reduces training instability in comparison to the commonly used GDA [56] and Best Reponse (BR) [12] methods. [17, 69] prove convergence of Stackelberg games under smooth player cost functions and show that they reduce the cycling behavior to find an equilibrium and allow for better convergence.

# 4 A Ranking Game for Imitation Learning

In this section, we first formalize the notion of the proposed two-player general-sum ranking game for imitation learning. We then propose a practical instantiation of the ranking game through a novel ranking-loss ($L_k$). The proposed ranking game gives us the flexibility to incorporate additional rankings—both auto-generated (a form of data augmentation mentioned as 'auto' in Fig. 1) and offline ('pref' in Fig. 1)—which improves learning efficiency. Finally, we discuss the Stackelberg formulation for the two-player ranking game and discuss two algorithms that naturally arise depending on which player is designated as the leader.

## 4.1 The Two-Player Ranking Game Formulation

We present a new framework, `rank-game`, for imitation learning which casts it as a general-sum *ranking game* between two players — a reward and a policy.

$$\underbrace{\text{argmax}_{\pi \in \Pi} J(R; \pi)}_{\text{Policy Agent}} \quad \underbrace{\text{argmin}_{R \in \mathcal{R}} L(\mathcal{D}^p; R)}_{\text{Reward Agent}}$$

In this formulation, the policy agent maximizes the reward by interacting with the environment, and the reward agent attempts to find a reward function that satisfies a set of pairwise behavior rankings in the given dataset $\mathcal{D}^p$; a reward function satisfies these rankings if $\mathbb{E}_{\rho^{\pi^i}}[R(s,a)] \le \mathbb{E}_{\rho^{\pi^j}}[R(s,a)]$, $\forall \rho^{\pi^i} \preceq \rho^{\pi^j} \in \mathcal{D}^p$, where $\rho^{\pi^i}, \rho^{\pi^j}$ can be state-action or state vistitations.

The dataset of pairwise behavior rankings $\mathcal{D}^p$ can be comprised of the implicit 'vanilla' rankings between the learning agent and the expert's policy behaviors ($\rho^\pi \preceq \rho^E$), giving us the classical IRL methods when a specific ranking loss function – *supremum-loss* is used [28, 22, 34]. If rankings are provided between trajectories, they can be reduced to the equivalent ranking between the corresponding state-action/state visitations. In the case when $\mathcal{D}^p$ comprises purely of offline trajectory performance rankings then, under a specific ranking loss function (*Luce-shepard*), the ranking game

---

**Algorithm 1** Meta algorithm: `rank-game` (vanilla) for imitation

---

1: Initialize policy $\pi_\theta^0$, reward funtion $R_\phi$, empty dataset $\mathcal{D}^\pi$. empirical expert data $\hat{\rho}^E$
2: **for** $t = 1..T$ iterations **do**
3:   Collect empirical visitation data $\hat{\rho}^{\pi_\theta^t}$ with $\pi_\theta^t$ in the environment. Set $\mathcal{D}^\pi = \{(\hat{\rho}^\pi \preceq \hat{\rho}^E)\}$
4:   Train reward $R_\phi$ to satisfy rankings in $\mathcal{D}^\pi$ using ranking loss $L_k$ in equation 3.
5:   Optimize policy under the reward function: $\pi_\theta^{t+1} \leftarrow \mathrm{argmax}_{\pi'} J(R_\phi; \pi')$
6: **end for**

---

reduces to prior reward inference methods like T-REX [10, 11, 9, 13]. Thus, the ranking game affords us a broader perspective of imitation learning, going beyond only using expert demonstrations.

### 4.2  Ranking Loss $L_k$ for the Reward Agent

We use a *ranking-loss* to train the reward function—an objective that minimizes the distortion [31] between the ground truth ranking for a pair of entities $\{x, y\}$ and rankings induced by a parameterized function $R : \mathcal{X} \to \mathbb{R}$ for a pair of scalars $\{R(x), R(y)\}$. One type of such a ranking-loss is the *supremum-loss* in the classical imitation learning setup.

We propose a class of ranking-loss functions $L_k$ that attempt to induce a performance gap of $k$ for all behavior preferences in the dataset. Formally, this can be implemented with the regression loss:

$$L_k(\mathcal{D}^p; R) = \mathbb{E}_{(\rho^{\pi^i}, \rho^{\pi^j}) \sim \mathcal{D}^p} \Big[ \mathbb{E}_{s,a \sim \rho^{\pi^i}} \big[ (R(s,a) - 0)^2 \big] + \mathbb{E}_{s,a \sim \rho^{\pi^j}} \big[ (R(s,a) - k)^2 \big] \Big]. \quad (3)$$

where $\mathcal{D}^p$ contains behavior pairs $(\rho^{\pi^i}, \rho^{\pi^j})$ s.t $\rho^{\pi^i} \preceq \rho^{\pi^j}$.

The proposed ranking loss allows for learning *bounded rewards with user-defined scale $k$* in the agent and the expert visitations as opposed to prior works in Adversarial Imitation Learning [28, 20, 22]. Reward scaling has been known to improve learning efficiency in deep RL; a large reward scale can make the optimization landscape less smooth [27, 24] and a small scale might make the action-gap small and increase susceptibility to extrapolation errors [7]. In contrast to the *supremum* loss, $L_k$ can also naturally incorporate rankings provided by additional sources by learning a reward function satisfying all specified pairwise preferences. The following theorem characterizes the equilibrium of the `rank-game` for imitation learning when $L_k$ is used as the ranking-loss.

**Theorem 4.1.** *(Performance of the `rank-game` equilibrium pair) Consider an equilibrium of the imitation `rank-game` $(\hat{\pi}, \hat{R})$, such that the ranking loss $L_k$ generalization error is bounded by $2R_{max}^2 \epsilon_r$ and the policy is near-optimal with $J(\hat{R}; \hat{\pi}) \geq J(\hat{R}; \pi) - \epsilon_\pi \ \forall \pi$, then at this equilibrium pair under the expert's unknown reward function $R_{gt}$ bounded in $[0, R_{max}^E]$:*

$$\big| J(R_{gt}, \pi^E) - J(R_{gt}, \hat{\pi}) \big| \leq \frac{4 R_{max}^E \sqrt{\frac{(1-\gamma)\epsilon_\pi + 4R_{max}\sqrt{\epsilon_r}}{k}}}{1 - \gamma} \quad (4)$$

*If reward is a state-only function and only expert observations are available, the same bound applies to the LfO setting.*

*Proof.* We defer the proof to Appendix A. □

**Theoretical properties:** We now discuss some theoretical properties of $L_k$. Theorem 1 shows that `rank-game` has an equilibrium with bounded performance gap with the expert. An optimization step by the policy player, under a reward function optimized by the reward player, is equivalent to minimizing an $f$-divergence with the expert. Equivalently, at iteration $t$ in Algorithm 1: $\max_{\pi^t} \mathbb{E}_{\rho^{\pi^t}}[R_t^*] - \mathbb{E}_{\rho^{\pi^E}}[R_t^*] = \min_{\pi^t} D_f(\rho^{\pi^t} \| \rho^{\pi^E})$. We elaborate on the regret of this idealized algorithm in Appendix A. Theorem 1 suggests that large values of $k$ can guarantee the agent's performance is close to the expert. In practice, we observe intermediate values of $k$ also preserve imitation equilibrium optimality with a benefit of promoting sample efficient learning (as an effect of reward scaling

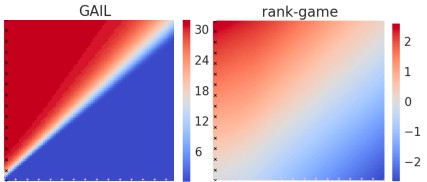

Figure 2: Figure shows learned reward function when agent and expert has a visitation shown by pink and black markers respectively. `rank-game` (auto) results in smooth reward functions more amenable to gradient-based policy optimization compared to GAIL.

described earlier). We discuss this observation further in Appendix D.9. `rank-game` naturally extends to the LfO regime under a state-only reward function where Theorem 4.1 results in a divergence bound between state-visitations of the expert and the agent. A state-only reward function is also a sufficient and necessary condition to ensure that we learn a dynamics-disentangled reward function [20].

$L_k$ can incorporate additional preferences that can help learn a regularized/shaped reward function that provides better guidance for policy optimization, reducing the exploration burden and increasing sample efficiency for IRL. A better-guided policy optimization is also expected to incur a lower $\epsilon_\pi$. However, augmenting the ranking dataset can lead to decrease in the intended performance gap $(k_{eff} < k)$ between the agent and the expert (Appendix A). This can loosen the bound in Eq 4 and lead to non-optimal imitation learning. We hypothesize that given informative preferences, decreased $\epsilon_\pi$ can compensate potentially decreased intended performance gap $k_{eff}$ to ensure near optimal imitation. In our experiments, we observe this hypothesis holds true; we enjoy sample efficiency benefits without losing any asymptotic performance. To leverage these benefits, we present two methods for augmenting the ranking dataset below and defer the implementation details to Appendix B.

### 4.2.1  Generating the Ranking Dataset

**Reward loss w/ automatically generated rankings (auto)**: In this method, we assume access to the behavior generating trajectories in the ranking dataset. For each pairwise comparison $\rho_i \preceq \rho_j$ present in the dataset, $L_k$ sets the regression targets for states in $\rho_i$ to be 0 and for states visited by $\rho_j$ to be $k$. Equivalently, we can rewrite minimizing $L_k$ as regressing an input of trajectory $\tau_i$ to vector $\mathbf{0}$, and $\tau_j$ to vector $k\mathbf{1}$ where $\tau_i, \tau_j$ are trajectories that generate the behavior $\rho_i, \rho_j$ respectively. We use the comparison $\rho_i \preceq \rho_j$ to generate additional behavior rankings $\rho_i \preceq \rho_{\lambda_1, ij} \preceq \rho_{\lambda_2, ij} .. \preceq \rho_{\lambda_P, ij} \preceq \rho_j$ where $0 < \lambda_1 < \lambda_2 < ... < \lambda_P < 1$. The behavior $\rho_{\lambda_p, ij}$ is obtained by independently sampling the trajectories that generate the behaviors $\rho_i, \rho_j$ and taking convex combinations i.e $\tau_{\lambda_p, ij} = \lambda_p \tau_i + (1 - \lambda_p)\tau_j$ and their corresponding reward regressions targets are given by $\lambda_p \mathbf{0} + (1 - \lambda_p)k\mathbf{1}$.

This form of data augmentation can be interpreted as mixup [68] regularization in the trajectory space. Mixup has been shown to improve generalization and adversarial robustness [25, 68] by regularizing the first and second order gradients of the parameterized function. Following the general principle of using a smoothed objective with respect to inputs to obtain effective gradient signals, explicit smoothing in the trajectory-space can also help reduce the policy optimization error $\epsilon_\pi$. A didactic example showing rewards learned using this method is shown in Figure 2. In a special case when the expert's unknown reward function is linear in observations, these rankings reflect the true underlying rankings of behaviors.

**Reward loss w/ offline annotated rankings (pref)**: Another way of increasing learning efficiency is augmenting the ranking dataset containing the vanilla ranking ($\rho^\pi \preceq \rho^E$) with offline annotated rankings. These rankings may be provided by a human observer or obtained using an offline dataset of behaviors with annotated reward information, similar to the datasets used in offline RL [19, 41]. We combine offline rankings by using a weighted loss between $L_k$ for satisfying vanilla rankings ($\rho^\pi \preceq \rho^E$) and offline rankings, grounded by an expert. Providing offline rankings alone that are sufficient to explain the reward function of the expert [10] is often a difficult task and the number of offline preferences required depends on the complexity of the environment. In the LfO setting, learning from an expert's state visitation alone can be a hard problem due to exploration requirements [36]. This ranking-loss combines the benefits of using preferences to shape the reward function and guide policy improvement while using the expert to guarantee near-optimal performance.

### 4.3  Optimizing the Two-Player General-Sum Ranking Game as a Stackelberg Game

Solving the ranking-game in the Stackelberg setup allows us to propose two different algorithms depending on which agent is set to be the leader and utilize the learning stability and efficiency afforded by the formulation as studied in [51, 69, 17].

**Policy as leader (PAL)**: Choosing policy as the leader implies the following optimization:

$$\max_\pi \left\{ J(\hat{R}; \pi) \ \ s.t. \ \ \hat{R} = \arg\min_R L(\mathcal{D}^\pi; R) \right\} \tag{5}$$

**Reward as leader (RAL):** Choosing reward as the leader implies the following optimization:

$$\min_{\hat{R}} \left\{ L(\mathcal{D}^\pi; \hat{R}) \;\; s.t \;\; \pi = \arg\max_\pi J(\hat{R}; \pi) \right\} \tag{6}$$

We follow the first order gradient approximation for leader's update from previous work [51] to develop practical algorithms. This strategy has been proven to be effective and avoids the computational complexity of calculating the implicit Jacobian term $(d\theta_B^*/d\theta_A)$. PAL updates the reward to near convergence on dataset $\mathcal{D}^\pi$ ($\mathcal{D}^\pi$ contains rankings generated using the current policy agent only $\pi \preceq \pi^E$) and takes a few policy steps. Note that even after the first-order approximation, this optimization strategy differs from GDA as often only a few iterations are used for training the reward even in hyperparameter studies like [46]. RAL updates the reward conservatively. This is achieved through aggregating the dataset of implicit rankings from all previous policies obtained during training. PAL's strategy of using on-policy data $\mathcal{D}^\pi$ for reward training resembles that of methods including GAIL [28, 62], $f$-MAX [22], and $f$-IRL [45]. RAL uses the entire history of agent visitation to update the reward function and resembles methods such as apprenticeship learning and DAC [1, 37]. PAL and RAL bring together two seemingly different algorithm classes under a unified Stackelberg game viewpoint.

# 5   Experimental Results

We compare `rank-game` against state-of-the-art LfO and LfD approaches on MuJoCo benchmarks having continuous state and action spaces. The LfO setting is more challenging since no actions are available, and is a crucial imitation learning problem that can be used in cases where action modalities differ between the expert and the agent, such as in robot learning. We focus on the LfO setting in this section and defer the LfD experiments to Appendix D.2. We denote the imitation learning algorithms that use the proposed ranking-loss $L_k$ from Section 4.2 as RANK-{PAL, RAL}. We refer to the `rank-game` variants which use automatically generated rankings and offline preferences as (auto) and (pref) respectively following Section 4.2. In all our methods, we rely on an off-policy model-free algorithm, Soft Actor-Critic (SAC) [26], for updating the policy agent.

We design experiments to answer the following questions:

1. *Asymptotic Performance and Sample Efficiency*: Is our method able to achieve near-expert performance given a limited number (1) of expert observations? Can our method learn using fewer environment interactions than prior state-of-the-art imitation learning (LfO) methods?

2. *Utility of preferences for imitation learning*: Current LfO methods struggle to solve a number of complex manipulation tasks with sparse success signals. Can we leverage offline annotated preferences through `rank-game` in such environments to achieve near-expert performance?

3. *Choosing between PAL and RAL methods*: Can we characterize the benefits and pitfalls of each method, and determine when one method is preferable over the other?

4. *Ablations for the method components*: Can we establish the importance of hyperparameters and design decisions in our experiments?

**Baselines:** We compare RANK-PAL and RANK-RAL against 6 representative LfO approaches that covers a spectrum of on-policy and off-policy model-free methods from prior work: GAIfO [62, 28], DACfO [37], BCO [61], $f$-IRL [45] and recently proposed OPOLO [71] and IQLearn [21]. We do not assume access to expert actions in this setting. Our LfD experiments compare to the IQLearn [21], DAC [37] and BC baselines. Detailed description for baselines can be found in Appendix D.2.

## 5.1   Asymptotic Performance and Sample Efficiency

In this section, we compare RANK-PAL(auto) and RANK-RAL(auto) to baselines on a set of MuJoCo locomotion tasks of varying complexities: `Swimmer-v2`, `Hopper-v2`, `HalfCheetah-v2`, `Walker2d-v2`, `Ant-v2` and `Humanoid-v2`. In this experiment, we provide one expert trajectory for all methods and do not assume access to any offline annotated rankings.

**Asymptotic Performance**: Table 2 shows that both `rank-game` methods are able to reach near-expert asymptotic performance with a single expert trajectory. BCO shows poor performance which

| Env | Hopper | HalfCheetah | Walker | Ant | Humanoid |
|---|---|---|---|---|---|
| BCO | 20.10±2.15 | 5.12±3.82 | 4.00±1.25 | 12.80±1.26 | 3.90±1.24 |
| GaIFO | 81.13± 9.99 | 13.54±7.24 | 83.83±2.55 | 20.10±24.41 | 3.93±1.81 |
| DACfO | 94.73±3.63 | 85.03±5.09 | 54.70±44.64 | 86.45±1.67 | 19.31±32.19 |
| $f$-IRL | 97.45± 0.61 | 96.06±4.63 | **101.16±1.25** | 71.18±19.80 | 77.93±6.372 |
| OPOLO | 89.56±5.46 | 88.92±3.20 | 79.19±24.35 | 93.37± 3.78 | 24.87±17.04 |
| RANK-PAL(ours) | 87.14± 16.14 | 94.05±3.59 | 93.88±0.72 | **98.93±1.83** | **96.84±3.28** |
| RANK-RAL(ours) | **99.34±0.20** | **101.14±7.45** | 93.24±1.25 | 93.21±2.98 | 94.45±4.13 |
| Expert | 100.00± 0 | 100.00± 0 | 100.00± 0 | 100.00± 0 | 100.00± 0 |
| $(|\mathcal{S}|, |\mathcal{A}|)$ | $(11, 3)$ | $(17, 6)$ | $(17, 6)$ | $(111, 8)$ | $(376, 17)$ |

Table 2: Asymptotic normalized performance of LfO methods at 2 million timesteps on MuJoCo locomotion tasks. The standard deviation is calculated with 5 different runs each averaging over 10 trajectory returns. For unnormalized score and more details, check Appendix D. We omit IQlearn due to poor performance.

can be attributed to the compounding error problem arising from its behavior cloning strategy. GAIfO and DACfO use GDA for optimization with a supremum loss and show high variance in their asymptotic performance whereas `rank-game` methods are more stable and low-variance.

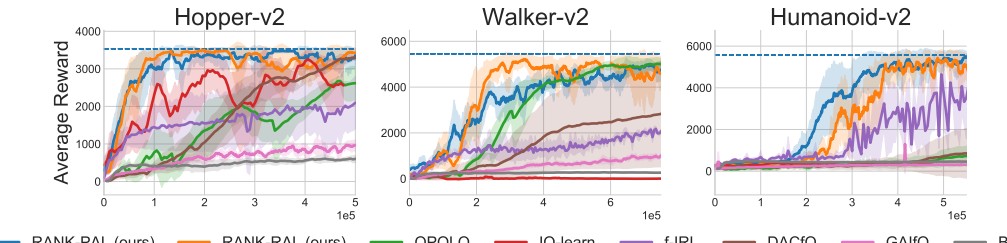

Figure 3: Comparison of performance on OpenAI gym benchmark tasks. The shaded region represents standard deviation across 5 random runs. RANK-PAL and RANK-RAL substantially outperform the baselines in sample efficiency. Complete set of results can be found in Appendix D.1

**Sample Efficiency**: Figure 3 shows that RANK-RAL and RANK-PAL are among the most sample efficient methods for the LfO setting, outperforming the recent state-of-the-art method OPOLO [71] by a significant margin. We notice that IQLearn fails to learn in the LfO setting. This experiment demonstrates the benefit of the combined improvements of the proposed ranking-loss with automatically generated rankings. Our method is also simpler to implement than OPOLO, as we require fewer lines of code changes on top of SAC and need to maintain fewer parameterized networks compared to OPOLO which requires an additional inverse action model to regularize learning.

## 5.2 Utility of Preferences in Imitation

Our experiments on complex manipulation environments—door opening with a parallel-jaw gripper [70] and pen manipulation with a dexterous adroit hand [50] – reveal that none of the prior LfO methods are able to imitate the expert even under increasing amounts of expert data. This failure of LfO methods can be potentially attributed to the exploration requirements of LfO compared to LfD [36], coupled with the sparse successes encountered

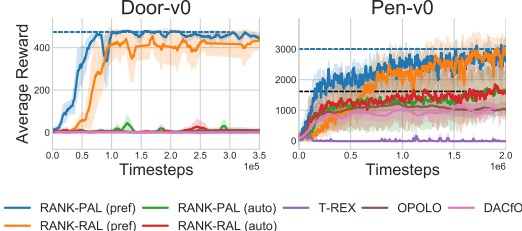

Figure 4: Offline annotated preferences can help solve LfO tasks in the complex manipulation environments Pen-v0 and Door, whereas prior LfO methods fail. Black dotted line shows asymptotic performance of RANK-PAL (auto) method.

in these tasks, leading to poorly guided policy gradients. In these experiments, we show that `rank-game` can incorporate additional information in the form of offline annotated rankings to guide the agent in solving such tasks. These offline rankings are obtained by uniformly sampling a small set of trajectories (10) from the replay buffer of SAC [26] labeled with a ground truth reward function. We use a weighted ranking loss (pref) from Section 4.2.

Figure 4 shows that RANK-PAL/RAL(pref) method leveraging offline ranking is the only method that can solve these tasks, whereas prior LfO methods and RANK-PAL/RAL(auto) with automatically

generated rankings struggle even after a large amount of training. We also point out that T-REX, a method that learns using the preferences alone is unable to achieve near-expert performance, thereby highlighting the benefits of learning from expert demonstrations alongside a set of offline preferences.

### 5.3 Comparing PAL and RAL

PAL uses the agent's current visitation for reward learning, whereas RAL learns a reward consistent with all rankings arising from the history of the agent's visitation. These properties can present certain benefits depending on the task setting. To test the potential benefits of PAL and RAL, we consider two non-stationary imitation learning problems, similar to [50] – one in which the expert changes it's intent and the other where dynamics of the environment change during training in the Hopper-v2 locomotion task. For changing intent, we present a new set of demonstrations where the hopper agent hops backwards rather than forward. For changing environment dynamics, we increase the mass of the hopper agent by a factor of 1.2. Changes are

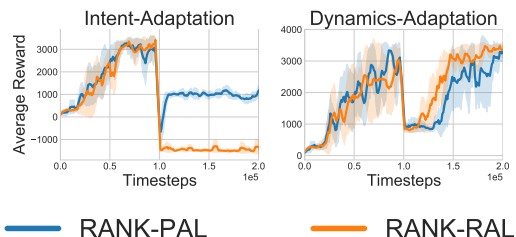

Figure 5: We compare the relative strengths of PAL and RAL. Left plot shows a comparison when the goal is changed, and right plot shows a comparison when dynamics of the environment is changed. These changes occur at 1e5 timesteps into training. PAL adapts faster to changing intent and RAL adapts faster to changing dynamics.

introduced at 1e5 time steps during training at which point we notice a sudden performance drop. In Figure 5 (left), we notice that PAL adapts faster to intent changes, whereas RAL needs to unlearn the rankings obtained from the agent's history and takes longer to adapt. Figure 5 (right) shows that RAL adapts faster to the changing dynamics of the system, as it has already learned a good global notion of the dynamics-disentangled reward function in the LfO setting, whereas PAL only has a local understanding of reward as a result of using ranking obtained only from the agent's current visitation.

**Ablation of Method Components:** Appendix D contains eight additional experiments to study the importance of hyperparameters and design decisions. Our ablations validate the importance of using automatically generated rankings, the benefit of ranking loss over *supremum* loss, and sensitivity to hyperparameters like the intended performance gap $k$, policy iterations, and the reward regularizer.

## 6 Conclusion

In this work, we present a new framework for imitation learning that treats imitation as a two-player ranking-game between a policy and a reward function. Unlike prior works in imitation learning, the ranking game allows incorporation of rankings over suboptimal behaviors to aid policy learning. We instantiate the ranking game by proposing a novel ranking loss which guarantees agent's performance to be close to expert for imitation learning. Our experiments on simulated MuJoCo tasks reveal that utilizing additional ranking through our proposed ranking loss leads to improved sample efficiency for imitation learning, outperforming prior methods by a significant margin and solving some tasks which were unsolvable by previous LfO methods.

**Limitations and Negative Societal Impacts:** Preferences obtained in real world are usually noisy [40, 32, 8] and one limitation of `rank-game` is that it does not suggest a way to handle noisy preferences. Second, `rank-game` proposes modifications to learn a reward function amenable to policy optimization but these hyperparameters are set manually. Future work can explore methods to automate learning such reward functions. Third, despite learning effective policies we observed that we do not learn reusable robust reward functions [45]. Negative Societal Impact: Imitation learning can cause harm if given demonstrations of harmful behaviors, either accidentally or purposefully. Furthermore, even when given high-quality demonstrations of desirable behaviors, our algorithm does not provide guarantees of performance, and thus could cause harm if used in high-stakes domains without sufficient safety checks on learned behaviors.

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
