# OpenReview forum: "A Ranking Game for Imitation Learning"
_NeurIPS.cc/2022/Conference — NeurIPS 2022 Submitted_

### Official Review · Reviewer_MNRH · 2022-07-08

**Rating:** 5
**Confidence:** 4
**Soundness:** 2 fair
**Presentation:** 3 good
**Contribution:** 2 fair

**Summary:**

The authors create a new method for learning a policy through IRL. The method contains two different agents: (i) a policy agent; and (ii) a reward agent. The authors perform a set of experiments with the OpenAI Gym + MuJoCo environments and show that they achieve better results than prior work.

**Questions:**

- How much time does it take to run each experiment in each of these methods? I think that a reason that the authors used so few trajectories in their experiments could be a hint that even though RANK is sample efficient, it does not seem to be time efficient. Clarification on time efficiency would be very important here.

**Limitations:**

- I think the authors addressed some of the concerns I had during my review in their limitation section. The ones they have not addressed are pointed out in the weakness section of this review.

**Strengths And Weaknesses:**

- Strengths:
    - The authors create a consistent work with a good explanation of why they created the method, and how the method works (theoretically and practically).
    - They perform extensive ablations on their method and how each part of the method reacts when deviating from the proposed method in the main material.
    - The work is well presented, with good writing and a couple of minor misspellings, nothing major that would damage the reading comprehension and experience.

- Weaknesses
    - I think that the major weakness of this work is the lack of experimentation with sub-optimal expert samples. With each year, we have more IL methods being developed and one of the main difficulties for a lot of them is to be resilient to sub-optimal samples. I think this should have been addressed in this work. Yes, the method is very efficient but what is the trade-off that controls this efficiency.
    - Even though RANK achieves good results with only one trajectory, I think there should be a study on how well it performs with more samples. We cannot forget that RANK has access to the environment and online samples have a higher cost than offline samples to collect and learn. I think this work would benefit from drawing this distinction and showing how much it gains when using different numbers of trajectories. Even more so when we think that there are other methods that were not used in these baselines (such as MobILE [1] and ILPO [2]) that already use a lower number of trajectories (10) while not having this iterative nature of RANK. Moreover, using a single trajectory was a limitation that RANK imposed to itself.
    - Table 2 only shows the Performance metric, and does not display the expert rewards, which hurts the readability of the results.
    - Another major concern that I have is the efficiency study. The authors say that their method achieves higher reward thresholds in 2M timesteps with only one trajectory (that can vary in sample size). However, this is not entirely true. If we consider that offline samples are more efficient to learn, we could argue that BCO, and all future work based on it, could be more efficient by using 10 trajectories and only needing a fraction of the timesteps in this experiment, for example.
    - Finally, regarding the experiments, 5 trajectories seem a lower number of trajectories to have an accurate statistical result. OPOLO used 10 times more samples to draw its average and standard deviation, while GaIFO used 100 trajectories for each experiment.
    - The authors’ overuse of math notations hurts the readability of their work quite extensively. Most of the time the iterator changed without proper explanation and it takes the reader quite some time to understand what the authors are iterating over.  I think that rewriting the equations and standardizing the notation would benefit this work.
    - If the policy agents interact with the environment to reduce the reward function, is it really IL? Why training an imitation learning agent if, in the end, we will have an agent trained by the reward function of the environment (which theoretically is the most optimal thing we could have in an MDP problem)?

    [1] Kidambi, Rahul, Jonathan Chang, and Wen Sun. "MobILE: Model-Based Imitation Learning From Observation Alone." *Advances in Neural Information Processing Systems*
     34 (2021): 28598-28611.

    [2] Edwards, Ashley, et al. "Imitating latent policies from observation." *International conference on machine learning*. PMLR, 2019.

---

> ### Author Response · Authors · 2022-08-02
> **Response to Reviewer MNRH (1/3)**
>
> Thank you for the detailed review of our paper and encouraging comments about the work. We are happy that you found our method well founded theoretically and practically, with clear writing, and ablation analysis to be extensive. We address your questions below. Please let us know if further clarification is needed.
>
> > I think that the major weakness of this work is the lack of experimentation with sub-optimal expert samples. With each year, we have more IL methods being developed and one of the main difficulties for a lot of them is to be resilient to sub-optimal samples. I think this should have been addressed in this work. Yes, the method is very efficient but what is the trade-off that controls this efficiency.
> >
> ****
> While learning from suboptimal experts is an important problem, it is not a focus of this work. We want to emphasize that we focus on an *equally important and complementary problem* of how to incorporate suboptimal data in the form of rankings to boost imitation learning. Progress in learning from optimal experts is still ongoing as evident in recent works [1, 2, 3]. We believe solving it is an important research direction and may influence future research in learning from suboptimal experts. Please let us know if we missed something in your comment.
>
> > Even though RANK achieves good results with only one trajectory, I think there should be a study on how well it performs with more samples. We cannot forget that RANK has access to the environment and online samples have a higher cost than offline samples to collect and learn. I think this work would benefit from drawing this distinction and showing how much it gains when using different numbers of trajectories. Even more so when we think that there are other methods that were not used in these baselines (such as MobILE [1] and ILPO [2]) that already use a lower number of trajectories (10) while not having this iterative nature of RANK. Moreover, using a single trajectory was a limitation that RANK imposed to itself.
> >
> ****
> We now also have *additional experiments* in the revised paper (Appendix Section D.11) that demonstrate how the performance of different methods scales with increasing expert offline data.
>
> **However, we would like to emphasize that this is a different setting from the primary focus of our work.** We highlight that learning from *minimal* expert demonstrations is an important problem that has been explored by a number of recent works in imitation [1,4]. Learning a policy from minimal expert demonstrations is also important when learning in the simulator, where the online interactions have no cost but collecting expert demonstrations is costly. This policy, trained on a simulator, can then be transferred to the real world. This is because directly training RL in the real world becomes infeasible due to RL’s sample complexity requirement (this approach can be seen in [5,6]).  Moreover, our approach accounts for reducing the sample requirements in online interactions compared to previous methods, which makes it a superior approach to deploy in the real world as well.
>
> In the paper, we present comparisons to the representative model-free approaches for imitation that have been shown to work well, including the state-of-the-art methods [1,3]. We point out that MobILE, despite being a model-based approach struggles to reach expert performance with 10 expert episodes, whereas our approach achieves near expert performance with just 1 expert trajectory (Hopper and Walker plots in Figure 2 in [7]). Second, we clarify that MobILE [7] is an iterative method, in contrast to what the reviewer points out.  ILPO is shown to be consistently outperformed by BCO in previous work [7] and hence we do not benchmark it.
>
> > Table 2 only shows the Performance metric, and does not display the expert rewards, which hurts the readability of the results.
> >
> ****
> We would like to clarify that the results reported in Table 2 are computed by normalizing the expert rewards to 100, so that performance in all environments can be compared on a similar scale. A table with unnormalized rewards can be found in Table 4 located in the appendix.

---

> > ### Author Response · Authors · 2022-08-02
> > **Response to Reviewer MNRH (2/3)**
> >
> > > Another major concern that I have is the efficiency study. The authors say that their method achieves higher reward thresholds in 2M timesteps with only one trajectory (that can vary in sample size). However, this is not entirely true. If we consider that offline samples are more efficient to learn, we could argue that BCO, and all future work based on it, could be more efficient by using 10 trajectories and only needing a fraction of the timesteps in this experiment, for example.
> > >
> > ****
> > We disagree with the speculation that BCO outperforms our method with more expert trajectories. *We have added additional experiments* with an increasing number of expert trajectories in Appendix D.11 . Our results demonstrate that even with an increased number of trajectories, our method RANK-GAME outperforms other methods.  Note that all of our expert trajectories are of a **fixed length of 1000 transitions**.
> >
> > We think a cause of confusion might be the belief that Behavior Cloning (BC) benefits from more expert data.  Note that we are considering *learning from observation* setting (BCO) where expert actions are unavailable, and finding expert actions that cause the next state is an exponentially harder problem compared to learning from expert demonstrations (state-*action*) as shown in [7]. BCO fails to scale gracefully even with increasing expert datasets of state-only data due to its compounding error problem coupled with poor exploration and has been known to suffer a quadratic in horizon regret [8]. In contrast, our method is based on the IRL formulation which suffers a linear in horizon regret.  It has also been empirically evident from previous work [1] that BCO fails to work well when we have a limited expert dataset.
> >
> > Second, we disagree that collecting online samples is *always* more costly than collecting expert demonstrations. RL has high sample complexity requirements to train on real-world and as a result, for applications like robotics, training in a simulator followed by sim2real is common practice. In such a simulated setting, *the online samples are not costly* and it can be quite hard to obtain expert samples (see [5,6]). Even when the online samples are more costly, our method demonstrates better sample efficiency as the experiments show (Section 5.1), and is a more promising candidate to deploy in the real world.
> >
> > > Finally, regarding the experiments, 5 trajectories seem a lower number of trajectories to have an accurate statistical result. OPOLO used 10 times more samples to draw its average and standard deviation, while GaIFO used 100 trajectories for each experiment.
> > >
> > ****
> > We apologize for the confusion. In Table 2, our results are averaged over 5 random seeds, and performance for each random seed is evaluated by averaging over 10 trajectories. i.e **our evaluation is an average of 50 trajectory returns, the same as OPOLO.**
> >
> > > The authors’ overuse of math notations hurts the readability of their work quite extensively. Most of the time the iterator changed without proper explanation and it takes the reader quite some time to understand what the authors are iterating over. I think that rewriting the equations and standardizing the notation would benefit this work.
> > >
> > ****
> > Thank you for helping to increase the readability of the work. We made the iterator more consistent in the revised paper.
> >
> > > If the policy agents interact with the environment to reduce the reward function, is it really IL? Why training an imitation learning agent if, in the end, we will have an agent trained by the reward function of the environment (which theoretically is the most optimal thing we could have in an MDP problem)?
> > >
> > ****
> > This appears to be a misunderstanding of our training and evaluation setting. In the settings considered in the paper, we do not assume access to ground truth reward. This is a natural setting that is common in robotics where we care about value alignment and robustness to reward misspecification and hence rely on imitation learning [9, 10] to learn from expert demonstrations/observations. We only assume access to the environment reward *for evaluation* as it provides a ground truth evaluation metric on how successfully we are able to imitate the expert. Please let us know if we misunderstood your comment and we would be happy to clarify.

---

> > > ### Author Response · Authors · 2022-08-02
> > > **Response to Reviewer MNRH (3/3)**
> > >
> > > Questions
> > >
> > > > How much time does it take to run each experiment in each of these methods? I think that a reason that the authors used so few trajectories in their experiments could be a hint that even though RANK is sample efficient, it does not seem to be time efficient. Clarification on time efficiency would be very important here
> > > >
> > > ****
> > > For the Hopper-v2 environment, RANK-GAME methods run ~2.67 X slower than a standard implementation of SAC [11] while being ~3X more sample efficient. In that sense, the run-time till convergence of RANK-GAME is comparable to a standard RL algorithm. In numbers, the training for RANK-PAL (auto) takes 0.0372 sec per environment timestep for Hopper-v2 environment. A complete run, as shown in the paper, of 500000 timesteps takes ~5.16 hours to complete.  This benchmarking is done on Intel(R) Xeon(R) CPU E5-2630 v4 @ 2.20GHz with a single NVIDIA TITAN V GPU. For comparison, SAC training on the same machine takes ~1.93 hours to complete.
> > >
> > > We highlight that our run time is independent of the number of expert trajectories available. As shown in Algorithm 2 in the Appendix, we update the reward using a fixed number of iterations ($n_{rew}$) that samples fixed size batches of the expert transitions. In many settings, real-world data is difficult/expensive to collect and in such situations, sample efficiency holds much more importance than the running time of the algorithm.
> > >
> > > We now also have additional experiments with more expert trajectories in Appendix D.11.
> > >
> > >
> > > References for the response:
> > >
> > > [1]: [Off-Policy Imitation Learning from Observations](https://papers.nips.cc/paper/2020/file/92977ae4d2ba21425a59afb269c2a14e-Paper.pdf), Zhu et al
> > >
> > > [2]: [Of Moments and Matching: A Game-Theoretic Framework for Closing the Imitation Gap](https://arxiv.org/pdf/2103.03236.pdf) - Swamy et al
> > >
> > > [3]: [IQ-Learn: Inverse soft-Q Learning for Imitation](https://arxiv.org/abs/2106.12142), Garg et al
> > >
> > > [4]: [f-IRL: Inverse Reinforcement Learning via State Marginal Matching](https://arxiv.org/abs/2011.04709) , Ni et al
> > >
> > > [5]: [Learning Agile Robotic Locomotion Skills by Imitating Animals](https://arxiv.org/abs/2004.00784) , Peng et al
> > >
> > > [6]: [ASE: Large-Scale Reusable Adversarial Skill Embeddings for Physically Simulated Characters](https://arxiv.org/abs/2205.01906), Peng et al
> > >
> > > [7]: [MobILE: Model-Based Imitation Learning From Observation Alone](https://arxiv.org/abs/2102.10769), Kidambi et al
> > >
> > > [8]: [Of Moments and Matching: A Game-Theoretic Framework for Closing the Imitation Gap](https://arxiv.org/pdf/2103.03236.pdf) - Swamy et al
> > >
> > > [9]: [AI Safety Gridworlds](https://arxiv.org/abs/1711.09883) , Leike et al
> > >
> > > [10]: [Value Alignment Verification](https://arxiv.org/abs/2012.01557), Brown et al
> > >
> > > [11]: [Spinning Up in Deep Reinforcement Learning](https://github.com/openai/spinningup), Achiam et al

---

### Official Review · Reviewer_Q6bL · 2022-07-10

**Rating:** 6
**Confidence:** 2
**Soundness:** 3 good
**Presentation:** 3 good
**Contribution:** 3 good

**Summary:**

The paper proposes to model imitation learning as a two-player general sum game, where one player, the "reward player", aims to rank a set of behaviors correctly, where the "policy player" aims to maximize the reward the "reward player" uses for ranking. This formulation unifies multiple prior methods for LfD and LfO. Then, the authors propose a novel way to solve this two player game as a Stackelberg game which leads to a novel imitation learning algorithm that outperforms prior methods in a series of simulated MuJoCo experiments. The proposed algorithm learns from demonstrations and preferences simultaneously which allows it to combine the benefit of both.

**Questions:**

What is the intuitive motivation for choosing the proposed ranking loss?

Do you consider any of the theoretical results apart from Thm. 4.1 a key contribution of your paper?

Does you assume all comparisons in the dataset are noise-free?

Could the proposed approach be extended to active learning by modifying what the reward agent does (i.e., it also chooses queries to make to a human)?


**Minor suggestions:**

- Capitalization:
   - "bradley-terry" -> "Bradley-Terry" (Table 1)
   - "jacobian" -> "Jacobian" (line 253)
   - In the references: "stackelberg" -> "Stackelberg", "Iq-learn" -> "IQ-learn", "soft-q" -> "soft-Q", etc.

- Also, make sure named paragraphs are consistently capitalized. For example "Ablation of method components" (line 350), but "Limitations and Negative Societal Impacts" (line 363)



**Limitations:**

If the paper indeed assumes noise-free preferences, this should be made clearer when setting up the problem. It would also be good to include as a dimension in Table 1, to provide a fair comparison.

**Strengths And Weaknesses:**

**Strengths**
- The two-player game framework is a conceptually elegant way of describing and unifying existing methods for IL. It is neat how the choice of different ranking loss functions yields different algorithms.
- The proposed algorithm is quite natural, and it is analyzed theoretically and empirically.
- The empirical evaluation is particularly thorough. The authors compare to various natural baselines on different tasks. They provide ablations of various components of their algorithm, providing the reader with a clear understanding of the performance of their algorithm in MuJoCo environments.

**Weaknesses**
- The paper does not provide a good intuition for the proposed ranking loss. After reading Sec. 4.2 I understood the choice that was made, but did not have a good understanding of why this is the right choice among many possible ranking losses.
- Parts of the paper could be written more clearly. For example, in Sec. 4.2 it would be natural to split it up in subsections discussing the ranking loss, the theoretical properties, and then the ways to generate the ranking dataset.
- I found it hard to understand the theoretical results following Theorem 4.1 without reading the Appendix.
- If I understand correctly, the method assumes all preferences that are provided to be noise-free, which limits the applicability of the method significantly.
- The experimental evaluation is limited to MuJoCo locomotion tasks, which can sometimes be too simple to draw general conclusions.

---

> ### Author Response · Authors · 2022-08-02
> **Response to Reviewer Q6bL (1/2)**
>
> Thank you for the detailed review of our paper and encouraging comments about the method and results. We are encouraged that you found the approach conceptually elegant and the empirical evaluation thorough. We address your questions below. Please let us know if further clarification is needed.
>
> > The paper does not provide a good intuition for the proposed ranking loss. After reading Sec. 4.2 I understood the choice that was made, but did not have a good understanding of why this is the right choice among many possible ranking losses.
> >
> ****
> Please find our response to the question here in point 1 of the General Response comment.
>
> > Parts of the paper could be written more clearly. For example, in Sec. 4.2 it would be natural to split it up in subsections discussing the ranking loss, the theoretical properties, and then the ways to generate the ranking dataset.
> >
> ****
> We update Sec 4.2 in the paper to increase the clarity of the section. Thank you for pointing this out!
>
> > Do you consider any of the theoretical results apart from Thm. 4.1 a key contribution of your paper?
> >
> ****
> Yes, we believe it is important for the proposed algorithm to theoretically perform well in idealized settings. A quick reference to the theoretical results in the Appendix can be found in Line 194-197 of the main paper.  The theoretical results apart from Thm 4.1 are contributions of the paper, but note that they deviate from the practical algorithm that we propose. For example, the regret guarantee in Lemma A.2 assumes that we have an RL oracle optimizer that does provably efficient exploration which practical deep RL methods like SAC seldom do.  We do believe these theoretical properties are ultimately important and it would be concerning if we did not have them.
>
> > If I understand correctly, the method assumes all preferences that are provided to be noise-free, which limits the applicability of the method significantly.
> >
> ****
> We have added additional experiments in Section D.12 to investigate how the method’s performance is affected by noisy preferences. Our experiments on the Door manipulation task show that our method is found to be robust to ~60% noise in preferences ([1] found 16% robustness in their method). One of the explanations for this observation is that the noise is only added to the offline preferences and the expert behavior still serves to behave as optimal.
>
> In this work, we assume that comparisons are noise free. This is mentioned in the limitation section of the main paper. One of the reasons for our noise-free assumption is that we use very few trajectory comparisons in the dataset (~10). These trajectories are uniformly spaced apart in their returns which makes their behavior easily distinguishable and their comparisons are unlikely to be affected by noise (eg. in the Boltzmann rational model). We think that the noisy comparison setting is an interesting avenue for future work that can be combined with our work as our contributions are orthogonal to a number of methods [1,5,6] that incorporate noisy preferences. For example, replacing the ranking loss we currently use with a noise-robust ranking loss is a possible direction for future work.
>
>
>
> > Could the proposed approach be extended to active learning by modifying what the reward agent does (i.e., it also chooses queries to make to a human)?
> >
> ****
> Yes, indeed! While the method does not require active queries, we believe additional online queries are just extra data points for learning a more robust reward function using the ranking loss.
>
> > The experimental evaluation is limited to MuJoCo locomotion tasks, which can sometimes be too simple to draw general conclusions.
> >
> ****
> We would like to bring to the reviewer's attention that in addition to MuJoCo locomotion tasks, we also test our approach on more challenging MuJoCo manipulation tasks. We agree with the reviewer that learning from expert *state-action* information (Learning from demonstrations) can be easier in MuJoCo but Learning from Observations Only *is still a challenging task* as identified in previous works [2] due to the additional exploration complexity involved. In Section 5.2, we show two challenging MuJoCo manipulations tasks that were unsolvable by prior methods which attest to the difficulty of LfO. In contrast to previous works [2, 3, 4], we perform extensive experiments covering LfO and LfD settings in both locomotion and manipulation environments.

---

> > ### Author Response · Authors · 2022-08-02
> > **Response to Reviewer Q6bL (2/2)**
> >
> > > Writing suggestions
> > >
> > ****
> > We thank the reviewer for pointing out the writing improvements. We have incorporated all the suggestions in the revision and this will surely help increase the readability of the paper.
> >
> >
> > References for this response:
> >
> > [1]: [Extrapolating Beyond Suboptimal Demonstrations via Inverse Reinforcement Learning from Observations](https://arxiv.org/pdf/1904.06387.pdf) - Brown et al
> >
> > [2]: [MobILE: Model-Based Imitation Learning From Observation Alone](https://arxiv.org/abs/2102.10769), Kidambi et al
> >
> > [3]: [Off-Policy Imitation Learning from Observations](https://papers.nips.cc/paper/2020/file/92977ae4d2ba21425a59afb269c2a14e-Paper.pdf), Zhu et al
> >
> > [4]: [Of Moments and Matching: A Game-Theoretic Framework for Closing the Imitation Gap](https://arxiv.org/pdf/2103.03236.pdf) - Swamy et al
> >
> > [5]: [LESS is More: Rethinking Probabilistic Models of Human Behavior](https://arxiv.org/pdf/2001.04465.pdf), Bobu et al
> >
> > [6]: [Reward-rational (implicit) choice: A unifying formalism for reward learning](https://papers.nips.cc/paper/2020/file/2f10c1578a0706e06b6d7db6f0b4a6af-Paper.pdf), Jeon et al

---

> > > ### Comment · Reviewer_Q6bL · 2022-08-06
> > > **Thanks for the response!**
> > >
> > > I acknowledge the authors' response to my review. The responses clarify some open questions and I appreciate the authors promising to update corresponding explanations in the paper. I think this should improve the paper quite a bit. But I think there are still some remaining limitations, in particular in the empirical evaluation and assuming noise-free preferences. Overall I maintain my evaluation of the paper and will keep arguing for "Weak Accept".

---

### Official Review · Reviewer_ZWmg · 2022-07-11

**Rating:** 7
**Confidence:** 3
**Soundness:** 3 good
**Presentation:** 3 good
**Contribution:** 3 good

**Summary:**

The paper proposes a ranking-game setting for imitation learning by using a ranking loss for the reward agent, and incorporating automatically generated rankings data augmentation as well as offline annotated rankings. The proposed method show improved performance and data efficiencies compare to state-of-the-art baselines on several benchmark tasks.

**Questions:**

May the author give some more explanation on the automatically generated rankings. Why would a linear combination of trajectories give a linear combined ranking?

**Limitations:**

The major limitations are described at the end of the paper, which I feel is acceptable.

**Strengths And Weaknesses:**

Strengths:
1. The relationship between the proposed setting with the existing IRL frameworks is well described.
1. The author gives a performance bound for the proposed ranking loss at equilibrium.
2. The experiments for the proposed method without offline preference show clear sample efficiency improvement compared to other methods.
3. The experiments for using offline preference give a significant performance improvement. While the improvement is expected but is still impressive.
4. The ablation study is thorough.

Weaknesses:
1. I am not quite convinced about the automatically generated rankings. Why would a linear combination of trajectories give a linear combined ranking?
2. The other limitations are described at the end of the paper, which I feel is acceptable.

---

> ### Author Response · Authors · 2022-08-02
> **Response to Reviewer ZWmg**
>
> Thank you for the detailed review of our paper and encouraging comments about the approach and results. We address your questions below.
>
> > May the author give some more explanation on the automatically generated rankings. Why would a linear combination of trajectories give a linear combined ranking?
> >
> ****
> Please find our response to the question here in point 2 of the General Response comment.
>
>
> Please let us know if further clarification is needed or if you have any other questions that we can address.

---

### Official Review · Reviewer_NadH · 2022-07-11

**Rating:** 6
**Confidence:** 3
**Soundness:** 3 good
**Presentation:** 3 good
**Contribution:** 3 good

**Summary:**

The paper considers the standard imitation learning problem by treating imitation as a two-player ranking-based game between a policy and a reward. A novel ranking loss is proposed which has the ability to learn from both expert demonstrations and preferences while keeping the learned reward bounded. The authors introduce two optimization methods based on Stackelberg games with different leaders. Experiments show that the two proposed algorithms work well in LfO settings and outperforms the baselines. In addition, by incorporating additional offline annotated rankings, the proposed method can work in more complex environments where other baselines fail. Finally, the pros and cons of the proposed two algorithms are also discussed.

**Questions:**

The questions are mainly discussed in the Weaknesses. Here are the selected questions that I think are most important:

- The intuition behind the proposed ranking loss (Eq 3).

- How to generate the ranking dataset in the setting of reward loss w/ automatically generated rankings? Will this cause some problem when combining the automatically generated rankings with the vanilla rankings?

- Can you provide some insights on the reason why the additional annotated is so useful?

**Limitations:**

The limitations and potential negative societal impact mentioned in the paper are adequately addressed. One suggestion: the authors can consider how the framework will work in the setting of learning from suboptimal demonstrations.

**Strengths And Weaknesses:**

**Strengths**

The idea of treating imitation as a two-player ranking-based game is novel, which encompasses both a kind of IRL methods and ranking-based methods. The proposed ranking loss learns a smoother reward function (Figure 2) compare with the IRL methods. The experimental results are strong, showing the efficacy of the proposed methods and the benefit of the additional annotated rankings. The comparison of the two proposed method is interesting. The writing of the paper is generally good.

**Weaknesses**

- The ranking loss Eq (3) is novel, but it may lack some intuition. The loss of GAIL is based on occupancy measure matching, and the loss of AIRL makes it possible to learn the ground-truth reward under some assumptions, and the loss of TREX is based on Luce-Shepard rule. However, the proposed loss lack some theoretical background and I would appreciate if the authors can provide additional intuitions behind this loss.

- Line 215: Here the behavior generating trajectories in the ranking dataset is assumed to be accessible. How do you generate the ranking dataset? Also, if you use auto magically generated rankings, does it mean that the demonstrations are not fully optimal? Will this cause some problem when combining the automatically generated rankings with the vanilla rankings, e.g., the learned policy may behave better than the low-ranked trajectories in the demonstrations, leading to some noisy ranking data?

- In Section 5.2, the results in Door-v0 show that RANK-PAL (pref) is the only method that can solve the problem. Can you provide some insight why the additional annotated rankings are better than the automatically generated rankings, and why the additional annotated rankings can reduce the exploration requirements of LfO (line 315) that much?

- In Section 5.2, the authors only show the performance of RANK-PAL. How does RANK-RAL work in these scenarios?

- For the writing:
  - Line 21: [Leaning from expert data (imitation learning) alone]: "data" here is a bit unclear here since actions are also data.
  - Line 25: [assumes no environment interaction]: The definition of "environment interaction" is unclear. I think you mean the environment interaction while learning the reward function, since DREX needs to collect interaction data while collecting noisy trajectories. In addition, are you suggesting no environment interaction is a bad thing here? Why is this a bad thing?
  - Line 57: [suboptimal rankings]: This is vague. What do you mean by saying rankings are "suboptimal"?
  - Line 135: a right parenthesis is missed.
  - Line 182: What is $\epsilon_r$?

---

> ### Author Response · Authors · 2022-08-02
> **Response to Reviewer NadH**
>
> Thank you for the detailed review of our paper and the motivating comments about the approach and results. We are encouraged that you found our work novel, well-written, and our experimental results to be strong. We address your questions below. Please let us know if further clarification is needed.
>
> > the proposed loss lack some theoretical background and I would appreciate if the authors can provide additional intuitions behind this loss
> >
> ****
> Our proposed ranking loss is shown in Theorem 4.1 to perform a divergence matching between the state-action/state occupancy of the agent and the expert. In line 195, we show that the ranking loss function minimizes an f-divergence between the current agent and expert’s visitation at any given iteration of the algorithm. Note that we match a different type of f-divergence (Appendix Line 652) than GAIL (Jensen-Shannon).  We discuss regret guarantees in Appendix A, lemma 2. A discussion on the intuition of the ranking loss can be found in General Response, point 1.
>
> > How do you generate the ranking dataset? Also, if you use automatically generated rankings, does it mean that the demonstrations are not fully optimal? Will this cause some problem when combining the automatically generated rankings with the vanilla rankings, e.g., the learned policy may behave better than the low-ranked trajectories in the demonstrations, leading to some noisy ranking data?
> >
> ****
>
> Please find our response to the questions here in point 2 of the General Response comment.
>
>
>
> > Can you provide some insight why the additional annotated rankings are better than the automatically generated rankings, and why the additional annotated rankings can reduce the exploration requirements of LfO (line 315) that much?
> >
> ****
>
> In hard exploration tasks, automatically generated rankings may not contain enough information for guiding policy updates since the rankings are obtained by convex combinations of state/state-action trajectories between vanilla rankings. Offline trajectories, on the other hand, are obtained from an offline dataset like D4RL where each trajectory is annotated by a human-designed reward function. These rewards are well shaped and the preferences generated from these reward functions can help infer/leak valuable information about the shaped reward, which in turn helps policy optimization. It has been observed in previous works [1] (Figure 1) that an algorithm like SAC can fail to solve the task without a well-shaped reward function.
>
> > In Section 5.2, the authors only show the performance of RANK-PAL. How does RANK-RAL work in these scenarios?
> >
> ****
> Thanks for pointing this out. We have added *RANK-RAL experiments* in section 5.2.
>
> > For the writing:
> >
> ****
> Thank you for pointing out the writing suggestions. We have made the suggested writing corrections and we hope they will increase the readability of the paper. Thank you for pointing out our error (Line 25) in the citation for learning from preferences without interactions. We wanted to cite B-REX[3] instead of D-REX[4] and this has been corrected now. We emphasize that TREX[5] can be understood as an instantiation of our approach when online interactions are not allowed and we do not suggest that the absence of environmental interaction is a bad thing. With online interactions, we can often get better guarantees in state-action visitation matching [2].
>
> [1]: [Self-Supervised Online Reward Shaping in Sparse-Reward Environments](https://arxiv.org/pdf/2103.04529.pdf) - Memarian et al
>
> [2]: [Of Moments and Matching: A Game-Theoretic Framework for Closing the Imitation Gap](https://arxiv.org/pdf/2103.03236.pdf) - Swamy et al
>
> [3]: [Safe Imitation Learning via Fast Bayesian Reward Inference from Preferences](https://arxiv.org/abs/2002.09089) - Brown et al
>
> [4]: [Better-than-Demonstrator Imitation Learning via Automatically-Ranked Demonstrations](http://proceedings.mlr.press/v100/brown20a/brown20a.pdf) - Brown et al
>
> [5]: [Extrapolating Beyond Suboptimal Demonstrations via Inverse Reinforcement Learning from Observations](https://arxiv.org/abs/1904.06387?context=cs) - Brown et al

---

> > ### Comment · Reviewer_NadH · 2022-08-07
> > **Thanks for the response**
> >
> > I'd like to thank the authors for the response to me and other reviewers. The response addresses my concerns and the paper has been improved. I am positive about this paper and will keep my original score.

---

### Official Review · Reviewer_48YC · 2022-07-16

**Rating:** 5
**Confidence:** 4
**Soundness:** 3 good
**Presentation:** 2 fair
**Contribution:** 3 good

**Summary:**

In this paper, the imitation learning (IL) problem, including learning from observation (LfO) and learning from demonstration (LfD), is studied. The paper proposes a new solution to IL which include three key aspects: (1) a new ranking loss is proposed for IL, which is capable to learn whenever the rankings of the trajectories are given or not; (2) a mixup-like data augmentation strategy is introduced to smooth the loss landscape; (3) the IL process is formulated into Stackelberg game, allowing the studied of the update frequencies of reward and policy functions. The proposed method is tested under benchmark continuous control tasks, showing performance improvement over existing IL approaches.



**Questions:**

1. Can you provide some explanations on why the proposed ranking loss is better than classical ranking losses, under the IL setting?

**Limitations:**

The limitations and negative societal impact are properly discussed.

**Strengths And Weaknesses:**

Strengths:

1. The paper proposes several interesting ideas for IL. In my view, the most significant ones are two-fold. One is the importance of smoothing the loss landscape, which can be achieved by the proposed loss and the mixup-like data augmentation. Another is the Stackelberg game formulation. The experiment of task changes in Section 5.3 is very interesting to me.

2. The overall experimental performance is impressive.

Weaknesses:
1. The proposed approach is somehow complicated. There are several components included, making it hard to identify their contributions to the performance improvement. Even though a number of ablation study results are reported, they only show that none of the components is useless. Their precise contributions, as well as underlying mechanisms, remain unclear.

2. It still remains unclear to me why the proposed ranking loss could outperform other classical ranking losses. As discussed in the paper, the loss is somehow like the one proposed in [1]. While in [1], since the algorithm is Q-learning, the choice of the reward function seems to have a less impact on the optimization objective.

3. I think conducting deeper studies following the direction in Section 5.3 would make the paper more interesting. The results in Section 5.3 seem to indicate that the PAL training is robust to changes in reward functions, while the RAL robust to changes in dynamics. I think these results show that one of the major advantages of formulating IL into Stackelberg games is the convenience of dealing with task changes.

Overall, I think the paper is more on proposing several useful techniques for IL without exploring the underlying mechanism deeply, which can be of good practical usefulness.

[1] SQIL: Imitation Learning via Reinforcement Learning with Sparse Rewards, ICLR'20.

---

> ### Author Response · Authors · 2022-08-02
> **Response to Reviewer 48YC**
>
> Thank you for the detailed review of our paper and encouraging comments about the approach and results. We are encouraged you found the ideas interesting and the experimental performance to be impressive. We address your questions below. Please let us know if further clarification is needed.
>
> > It still remains unclear to me why the proposed ranking loss could outperform other classical ranking losses. As discussed in the paper, the loss is somehow like the one proposed in [1]. While in [1], since the algorithm is Q-learning, the choice of the reward function seems to have a less impact on the optimization objective
> >
> ****
> Please find our response to the first question here in point 1 of the General Response comment.
>
> For the second question, we emphasize that our proposed ranking loss **is only similar in motivation** **but differs significantly** from the one proposed in [1]. [1] sets the reward of the expert state-action empirical visitation to be 1 and agent’s to be 0. This approach does not use learned rewards and lacks theoretical correctness in the view of state-action occupancy matching [2]. The empirical evidence verifying that SQIL does not work as well compared to methods that use learned rewards can be found in [3,4].
>
> > I think conducting deeper studies following the direction in Section 5.3 would make the paper more interesting. I think these results show that one of the major advantages of formulating IL into Stackelberg games is the convenience of dealing with task changes
> >
>
> ****
> Thank you for pointing out the benefits offered by our proposed approach that frames IL in the Stackelberg formulation. We highlight that we include two experiments on task change and intent change to validate the benefit of this setting. While to our knowledge, we are the first to adapt IL in the Stackelberg setting, the *general* properties of the Stackelberg games for adaptation in non-stationary settings have been explored in previous work [5]. We consider a thorough empirical analysis of benefits from general Stackelberg games to be out of the scope of the current work.
>
> > The proposed approach is somehow complicated. There are several components included, making it hard to identify their contributions to the performance improvement. Even though a number of ablation study results are reported, they only show that none of the components is useless. Their precise contributions, as well as underlying mechanisms, remain unclear.
> >
> ****
> We include a discussion in the appendix section D.10 in the revised paper to explain the gains offered from different components for more clarity.
>
> We highlight that our ablations not only show that each component is beneficial but also show a detailed *quantitative* analysis of each component in the Appendix (Figure 8, 9, 10, 11, 16, and Table 3). The primary components that affect the empirical performance of the method are the ranking loss, range of reward function values, and reward-policy update frequency under the Stackelberg framework. The quantitative comparisons of removing these components can be found in Figure 11 and Figure 16 in the appendix. We note that, as can be seen in the ablation experiments, the other components (reward regularization [Appendix D.9], parameterized reward shaping [Appendix D.5]) contribute to a small improvement in performance. We also point out that one set of hyperparameters works well across all the tasks considered in the paper, making the method robust to environment variations.
>
> [1]: [SQIL: Imitation Learning via Reinforcement Learning with Sparse Rewards](https://arxiv.org/abs/1905.11108), Reddy et al
>
> [2]: [Energy-Based Imitation Learning](https://arxiv.org/pdf/2004.09395.pdf), Liu et al
>
> [3]: [Offline Learning from Demonstrations and Unlabeled Experience](https://arxiv.org/pdf/2011.13885.pdf) , Zolna et al
>
> [4]: [Semi-supervised reward learning for offline reinforcement learning](https://arxiv.org/pdf/2012.06899.pdf), Konyushkova et al
>
> [5]: [A Game Theoretic Framework for Model Based Reinforcement Learning](https://arxiv.org/abs/2004.07804) , Rajeswaran et al

---

### Author Response · Authors · 2022-08-02
**General response to the reviewers (1/2)**

We thank the reviewers for their insightful and positive feedback! We are encouraged that they found our work to present *several interesting ideas for Imitation Learning* (**48YC**), found the *ideas novel, conceptually elegant and sound, both theoretically and practically* (**NadH, ZWmg, Q6bL, MNRH**), and *well situated among related work* (**ZWmg, Q6bL**). We are also glad that they found *our experimental evaluation along with the ablation to be thorough* (**ZWmg, Q6bL**) and *the experimental performance to be impressive* (**48YC, NadH, ZWmg, Q6bL, MNRH**).

The two common questions raised by the reviewers were (1) the intuition behind our proposed ranking loss and (2) clarifications related to automatically generated rankings. We address these questions below.  We address the other questions and concerns from the reviewer’s comments individually and incorporate all the feedback in the paper revision.

1. **Intuition behind the choice of the ranking loss**

Utilizing preferences for reward inference has several advantages: (a) inferring a total order over behaviors, (b) reducing the exploration burden for policy learning, and (c) being easy to obtain. On the other hand, expert demonstrations can be hard to collect but contain crucial information about optimal agent behavior. Algorithms for preference-based reward inference and imitation/IRL from expert demonstrations have evolved separately and our aim is to unify them under the framework of a ranking game to take advantage of both modalities of information. An important component of a ranking game is the ranking loss function which should be able to learn from expert demonstrations as well as offline preferences while preserving theoretical guarantees of imitation learning.

Our proposed ranking loss learns a reward function that attempts to induce a user-defined performance gap between pairs of rankings. Minimizing this loss is characterized by a theoretical guarantee of f-divergence minimization with the expert’s state/state-action visitation for imitation learning. Note that we match a different type of f-divergence (Appendix Line 652) than GAIL (Jensen-Shannon). In line 195, we point out that the proposed ranking loss function minimizes an f-divergence between the current agent and expert’s visitation at any given iteration of the algorithm. We also discuss regret guarantees of the rank-game algorithm in Appendix A, lemma 2.

**Properties of the ranking loss:** The proposed ranking loss provides ease of policy optimization by controlling scale of the learned *bounded* reward function [6,7] along with learning a regularized reward function that is able to better guide policy optimization. The ability to incorporate preferences also helps infer a well-shaped reward function. This is in contrast to previous methods in Adversarial IL [8,9,10] where reward functions are often unbounded, non-smooth, and unable to leverage preferences between suboptimal trajectories.

**Why not other ranking losses?** : There are other classes of ranking loss, for instance, the large class of Lovasz-Bregman divergences [1] that have been useful in web ranking and clustering, but due to its discontinuous/non-smooth loss landscape, it worked poorly in practice with our gradient-based optimization setup. The current choice of ranking loss also helps us to control the amount of performance separation we want to enforce between each preference in contrast to other losses like in TREX [2], [3] which maximize this separation. This property is leveraged in our work (Line 789 and Appendix D.5) and it can be generalized in the future to give more weight to preferences that are more important than others. An interesting future direction is also to consider ranking losses that are robust to noisy preferences.

---

> ### Author Response · Authors · 2022-08-02
> **General response to the reviewers (2/2)**
>
> 2. **Clarification for automatically generated rankings**
>
> Automatically-generated ranking is a data-augmentation technique we leverage that regularizes the learned reward function. Before applying data augmentation, the ranking data in imitation consists of a behavior preference between the current policy and the expert policy (vanilla rankings).  The automatically generated rankings are interpolations generated using convex combinations of state/state-action trajectories from the vanilla rankings $\lambda [s_0^{agent}s_1^{agent}..s_H^{agent}]+ (1-\lambda) [s_0^{expert}s_1^{expert}..s_H^{expert}]$
>
> Key properties:
>
> 1. The ranking assignment (Line 222) for the interpolated trajectory is consistent with scalar value $\lambda$   and ensures that none of the generated trajectories are given a higher preference than the expert, ensuring that the expert is treated as being optimal.
> 2. Although this process can sometimes result in noisy preferences among the lower ranked trajectories, we emphasize (Line 226) that this method is a form of mixup regularization in trajectory space which improves generalization and adversarial robustness for reward training. Mixup is a regularization technique used in supervised learning, commonly for classification and regression tasks, where the convex combination of inputs is regressed to the corresponding convex combination of their labels. This procedure has been found to be successful both empirically[4] and theoretically[5].
>
> [1]: [The Lovasz-Bregman Divergence and connections to rank aggregation, clustering, and web ranking](https://arxiv.org/abs/1308.5275) - Iyer et al
>
> [2]: [Extrapolating Beyond Suboptimal Demonstrations via Inverse Reinforcement Learning from Observations](https://arxiv.org/pdf/1904.06387.pdf) - Brown et al
>
> [3]: [Deep Reinforcement Learning from Human Preferences](https://arxiv.org/pdf/1706.03741.pdf) - Christiano et al
>
> [4]: [mixup: Beyond Empirical Risk Minimization](https://arxiv.org/abs/1710.09412) - Zhang et al
>
> [5]: [How Does Mixup Help With Robustness and Generalization](https://arxiv.org/abs/2010.04819)? , Zhang et al
>
> [6]: [Increasing the action gap: New operators for reinforcement learning](https://arxiv.org/abs/1512.04860), Bellemare et al
>
> [7]: [Deep reinforcement learning that matters](https://arxiv.org/abs/1709.06560), Henderson et al
>
> [8]: [Generative Adversarial Imitation Learning](https://arxiv.org/abs/1606.03476), Ho et al
>
> [9]: [A Divergence Minimization Perspective on Imitation Learning Methods](https://arxiv.org/abs/1911.02256), Ghasemipour et al
>
> [10]: [Learning Robust Rewards with Adversarial Inverse Reinforcement Learning](https://arxiv.org/abs/1710.11248), Fu et al

---

### Author Response · Authors · 2022-08-09
**Follow-up response to the reviewers**

We hope the responses below answer the questions and concerns raised in the reviews. Since the author-reviewer discussion period is about to end soon, please let us know if there are any follow-up questions for the paper.

We hope that our explanation resolves your concerns, and would appreciate it if you would consider reevaluating your scores in light of our response. Thank you.

---

### Meta-Review · Area_Chair_9p6t · 2022-08-26

**Recommendation:** Reject
**Confidence:** Certain

**Metareview:**

I went through the paper, reviews and responses. This is a borderline paper. The negative to neutral reviews are more detailed and convincing.

**Award:**

No

---

### Decision · Program_Chairs · 2022-09-14

Reject